# Deep Learning for the Prediction of the Survival of Midline Diffuse Glioma with an H3K27M Alteration

**DOI:** 10.3390/brainsci13101483

**Published:** 2023-10-19

**Authors:** Bowen Huang, Tengyun Chen, Yuekang Zhang, Qing Mao, Yan Ju, Yanhui Liu, Xiang Wang, Qiang Li, Yinjie Lei, Yanming Ren

**Affiliations:** 1Department of Neurosurgery, West China Hospital of Sichuan University, No. 37, Guoxue Alley, Chengdu 610041, China; bowenhuang666@foxmail.com (B.H.); tychen2718@163.com (T.C.); 2012zykyx@sina.cn (Y.Z.); r89738759@126.com (Q.M.); juyanwestchina@126.com (Y.J.); h5369843756@163.com (Y.L.); k623768946@126.com (X.W.); w6765384579@163.com (Q.L.); 2College of Electronics and Information Engineering, Sichuan University, Chengdu 610065, China; l579scu2012@yeah.net

**Keywords:** diffuse midline glioma, H3K27M alteration, machine learning, DeepSurv, survival model

## Abstract

Background: The prognosis of diffuse midline glioma (DMG) patients with H3K27M (H3K27M-DMG) alterations is poor; however, a model that encourages accurate prediction of prognosis for such lesions on an individual basis remains elusive. We aimed to construct an H3K27M-DMG survival model based on DeepSurv to predict patient prognosis. Methods: Patients recruited from a single center were used for model training, and patients recruited from another center were used for external validation. Univariate and multivariate Cox regression analyses were used to select features. Four machine learning models were constructed, and the consistency index (C-index) and integrated Brier score (IBS) were calculated. We used the receiver operating characteristic curve (ROC) and area under the receiver operating characteristic (AUC) curve to assess the accuracy of predicting 6-month, 12-month, 18-month and 24-month survival rates. A heatmap of feature importance was used to explain the results of the four models. Results: We recruited 113 patients in the training set and 23 patients in the test set. We included tumor size, tumor location, Karnofsky Performance Scale (KPS) score, enhancement, radiotherapy, and chemotherapy for model training. The accuracy of DeepSurv prediction is highest among the four models, with C-indexes of 0.862 and 0.811 in the training and external test sets, respectively. The DeepSurv model had the highest AUC values at 6 months, 12 months, 18 months and 24 months, which were 0.970 (0.919–1), 0.950 (0.877–1), 0.939 (0.845–1), and 0.875 (0.690–1), respectively. We designed an interactive interface to more intuitively display the survival probability prediction results provided by the DeepSurv model. Conclusion: The DeepSurv model outperforms traditional machine learning models in terms of prediction accuracy and robustness, and it can also provide personalized treatment recommendations for patients. The DeepSurv model may provide decision-making assistance for patients in formulating treatment plans in the future.

## 1. Introduction

H3K27M-mutant diffuse midline glioma (H3K27M-DMG) is a new class in the 2016 WHO classification of central nervous system (CNS) [1], and H3K27M mutation was replaced by H3K27M alteration in the 2021 WHO classification of CNS tumors (fifth edition) [2]. Molecular analyses of biological tissue have revealed that the vast majority of these tumors possess a histone H3 gene mutation that most commonly occurs at either H3.1 (HIST1H3B/C) or H3.3 (H3F3A) and results in a H3K27M mutation (histone 3 lysine substitution for methionine at site 27 (H3K27M)) [3,4]. It behaves like a much more aggressive tumor if H3K27M is mutated and is classified as WHO Grade IV irrespective of histology [5]. This type of tumor not only grows in the brainstem, but also originates in other midline structures, such as the thalamus, gangliocapsular region, cerebellum, cerebellar peduncles, third ventricle, hypothalamus, and pineal region, as well as in the spinal cord [6]. H3K27M-DMG is the second most common childhood malignant brain tumor, with an incidence of 200–300 cases annually in the United States [7].

The prognosis of DMG patients with H3K27M alterations is poor, with a median survival period reported in the literature between 8.76 and 22.8 months [8,9,10]. The 2-year survival rate is less than 10% for patients receiving standard treatment, including surgery and adjuvant chemoradiation [11]. Currently, studies have shown that ATRX gene mutation, age, and radiation therapy are independent prognostic factors for H3K27M-DMG patients [12,13]. To date, however, a model that encourages accurate prediction of prognosis for such lesions on an individual basis remains elusive.

Traditional machine learning and deep learning constitute artificial intelligence, which is currently widely used in the medical field [14,15]. Zhou et al. [16] conducted survival analysis on intrahepatic cholangiocarcinoma using multicenter data based on machine learning modeling and prediction and achieved good results. In 2018, Katzman et al. [17] proposed DeepSurv, which is a survival prediction method based on neural networks. It can discover nonlinear relationships between different factors that traditional machine learning has difficulty detecting, and they further proved that the DeepSurv model is superior to traditional machine learning models in fitting patient survival and recommending treatment and has better predictive performance for complex survival data. Adeoye et al. [18] constructed machine learning models and a DeepSurv model to predict the malignant transformation-free survival of oral potentially malignant disorders. They found that, compared to traditional survival models, the DeepSurv model has the highest prediction accuracy and robustness. Individualized survival prediction for diseases can provide assistance for patients in making subsequent treatment decisions. However, there is currently no survival prediction based on deep learning for H3K27M-DMGs. Therefore, we aimed to construct an H3K27M-DMG survival model based on DeepSurv to predict patient prognosis, with the hope of guiding the individualized treatment of patients in clinical practice.

## 2. Materials and Methods

### 2.1. Patients and Definitions

We retrospectively enrolled diffuse midline glioma (DMG) patients from February 2016 to April 2022 at West China Hospital of Sichuan University. Patients who met the following inclusion and exclusion criteria were collected as a training set in this study. The inclusion criteria were as follows: craniotomy or biopsy surgery was performed at West China Hospital of Sichuan University from February 2016 to April 2022, and the postoperative pathological results showed H3K27M alteration. The exclusion criteria were as follows: (1) previous history of other craniocerebral operations; (2) the tumor was located in the spinal cord; (3) the tumor was located in the nonmidline region; and (4) loss at follow-up. H3K27M status was determined in patients by pyrosequencing analysis for H3F3A or HIST1H3B mutations. Patients meeting the same inclusion and exclusion criteria from Chengdu Shangjin Nanfu Hospital from February 2016 to April 2022 were selected as the external test set. All operations were performed by senior neurosurgery professors. This study was approved by the West China Hospital Ethics Committee, and written informed consent was exempt from the present study as a retrospective clinical study. We started accessing data for research purposes on 1 August 2022.

We collected the following information from patients based on medical history, surgical records, imaging records, pathological reports, telephone calls and outpatient follow-up for model training: age at diagnosis (0–100), sex (male or female), preoperative Karnofsky Performance Scale (KPS) score (0–100), tumor location (thalamus, midbrain, pontine, medulla, and basal ganglia), enhancement (enhancement in the T1 enhanced sequence of magnetic resonance imaging), tumor size (≥1 mm, ≥2 mm, ≥3 mm, ≥4 mm), and the extent of resection (resections were defined as gross total resection/GTR when 100% of the tumor was removed; subtotal resection/STR and partial resection/PR designate the tumoral remnant as <10% and <50%, respectively), adjuvant therapy (radiotherapy or chemotherapy treatment), ATRX expression (lost or intact), p53 positivity (positive or negative), Ki-67 level, MGMT status (unmethylated or methylated), survival/follow-up time in months, and survival status (dead or survived).

### 2.2. Feature Selection

We used univariate Cox regression analysis to screen for significant prognostic factors and included these characteristics in the multivariate Cox regression survival analysis. We use the cor function in R software to calculate the interrelationships between these features and test whether there is collinearity between them. When Pearson’s correlation value is ≥0.7, it means that these factors have a high degree of collinearity.

### 2.3. Model Construction of Machine Learning

We constructed four machine learning prediction models, including two traditional machine learning survival prediction models and two deep learning neural network models, and compared which of the four prediction models best fit the survival state of H3K27M-DMG. Traditional machine learning models include the Cox proportional hazard (CoxPH) model and Random Survival Forest (RSF) model. The deep neural network model includes the Neural Multi-Task Logistic Regression (N-MTLR) and DeepSurv [17] models (Figure 1). Research [19] has shown that the N-MTLR model is significantly better than that of the traditional survival prediction model in survival prediction. DeepSurv is a deep feedforward neural network that can predict the survival state of patients by using patient covariates. It includes an input layer, an intermediate hidden layer, and an output layer. We input the patient’s covariates into the neural network as input layers, with the middle layer consisting of fully connected neural nodes followed by a dropout layer. The model constantly automatically adjusts the feature weight and outputs the patient’s survival probability from the final output layer. We normalized KPS and one-hot encoded locations and then fed those covariables into the neural network.

### 2.4. Model Training

In the training of the model, we divided the data into a training set and a validation set according to the ratio of 75% to 25%; that is, all the training of the model was carried out in 75% of the datasets, and all the validation of the model was carried out in 25% of the dataset. All models were trained with 600 epochs and verified by 5-fold cross-validation, and their performance was tested by external test sets after training. When training the DeepSurv and N-MTLR models, we used a random hyperparameter search to optimize the model’s hidden layer number, neural nodes, activation function, dropout, optimizer, and iteration times.

### 2.5. Model Performance Measures

We calculated the consistency index (C-index) of the model in the training and testing sets. Time dependent areas under the receiver operating characteristic (ROC) curve (AUCs) were also calculated to evaluate machine learning models at 6, 12, 18, and 24 months. The closer the ROC curve is to the upper left corner, the better the predictive performance of the model (the larger the AUC value is, the higher the prediction accuracy of the model). Brier scores represent the mean square deviation between the observed patient status and the predicted survival probability. Its value is always between 0 and 1, and the closer the score is to 0, the better. Conversely, the higher the score is, the worse the prediction result and the worse the calibration level. Models with Brier scores less than 0.25 are considered useful in practice. To determine the overall performance of the model over all periods, we also calculated the integrated Brier score (IBS). We use the streamlit package (1.20.0) to create an app that can directly use the trained model to output the survival probability of patients at different times using patient information.

### 2.6. Statistical Analysis

SPSS software version 25.0 (IBM Corp., Armonk, NY, USA) was used for data analysis. Classified variables are described by percentages, and continuous variables are described by means ± standard deviations. Categorical variables were compared with the chi-squared test. Continuous variables were compared with independent-samples t tests or rank-sum tests. During this analysis process, all statistical tests were 2-sided, and a *p* value < 0.05 was considered statistically significant. The PySurvival package (version 0.2.1) in Python software (version 3.7.16) was used to build models. We used R version 4.1.0 for plotting. Kaplan–Meier analysis and log-rank testing were then performed using the Python lifelines survival analysis module.

## 3. Results

### 3.1. Patient Characteristics

We recruited 113 patients in the training set and 23 patients in the test set (Figure 2). The comparison of clinical information between the two groups is shown in Appendix A. After statistical comparison, patients in the training set were younger than those in the test set, with a significant difference (23.04 ± 16.84 years vs. 34.39 ± 17.27 years, *p* = 0.004). In terms of postoperative adjuvant radiotherapy, patients in the test set were more active than those in the training set. A total of 52.2% of patients chose radiotherapy, while only 27.4% chose radiotherapy in the training set, showing a statistically significant difference (*p* = 0.020). However, in terms of chemotherapy, there was no significant differences between the test set and training set (39.1% vs. 28.3%, *p* = 0.303). ATRT expression in the training set was significantly higher than that in the test set (68.1% vs. 52.1%, *p* = 0.022). The survival curves of the two groups are shown in Appendix A, and the results show no significant difference (*p* = 0.49). There were no statistically significant differences in sex, tumor location, surgical resection scope, preoperative KPS, or survival status between the two groups.

### 3.2. Feature Screening Results

Univariate Cox analysis (Table 1) showed a significant negative correlation between tumor size, enhancement, Ki67 expression, and patient survival time. The location of the tumor in the midbrain (HR; 0.733, 95% CI; 0.103–0.723) was a protective factor compared to the location of the tumor in the medulla. KPS (HR; 0.964, 95% CI; 0.952–0.975, *p* < 0.05), radiotherapy (HR; 0.203, 95% CI; 0.121–0.342, *p* < 0.05) and chemotherapy (HR; 0.240, 95% CI; 0.146–0.395, *p* < 0.05) showed a significant positive correlation with patient survival time. After multivariate Cox analysis, Ki67 expression lost significance (HR; 2.533, 95% CI; 0.511–12.565, *p* = 0.255). The results of collinearity analysis showed that the highest correlation coefficient was between radiotherapy and chemotherapy (Figure 3), with a value of 0.54 < 0.70. Finally, we included tumor size, tumor location, KPS, enhancement, radiotherapy, and chemotherapy for model training. Loss convergence graphs for the N-MTLR and DeepSurv models are shown in Appendix A.

### 3.3. Model Performance

The performance of the models in the two datasets is summarized in Table 2. The results show that the model can effectively fit and predict the survival status of patients in both the training and testing sets. The hyperparameter search results of the models are shown in Appendix A. In the training set, the DeepSurv model had the highest accuracy, with a c-index of 0.862, while the other three models had c-indexes of 0.819 (CoxPH), 0.824 (N-MTLR), and 0.845 (RSF). The prediction accuracy of the DeepSurv model in the test set decreased by only 0.051 compared to the training set, which is the least reduction among the four models. We plotted the IBS curves for each model, as shown in Appendix A. The IBS of all four models is less than 0.25, with the DeepSurv model having the lowest IBS of 0.093 among the four models. The ROC curves and AUC values of the four models at 6 months, 12 months, 18 months, and 24 months are shown in Figure 4. The AUC values of the four models gradually decrease with increasing time. The possible reason is that the number of patients who die increases over time, resulting in less data for subsequent model training, resulting in a decline in the performance of the model. The DeepSurv model has the highest AUC values at the four time nodes, which are 0.970 (0.919–1), 0.950 (0.877–1), 0.939 (0.845–1), and 0.875 (0.690–1). The best performance in the test set is still the DeepSurv model, which has a C-index and IBS of 0.811 and 0.147, respectively. The AUC values of this model at the four time points are 0.893 (0.827–0.972), 0.869 (0.782–0.961), 0.866 (0.776–0.962), and 0.803 (0.667–1), respectively.

### 3.4. Model Visualization

A heatmap of feature importance (Figure 5) shows the degree to which these features have an impact on the model when predicting patient prognosis. The results showed that the features that had the greatest impact on the DeepSurv, N-MTLR, and RSF models were KPS, tumor size, and KPS, respectively. We designed an interactive interface to more intuitively display the survival probability prediction results provided by the DeepSurv model. The surgeon inputs the prognosis information of the patient on the left side, and the survival probability of the patient at different times is automatically predicted immediately on the right side. This program can also visually compare the survival curves of patients after using different combinations of adjuvant radiotherapy and chemotherapy methods to select the treatment method that can most prolong the patient’s life. The visualization of the application’s functionality and output is shown in Appendix A.

## 4. Discussion

Survival analysis of the statistical model finds its application widely in clinical oncology in providing the prognosis of the disease to the patients by finding the probability that a patient would survive more than a particular time [20]. In the past, many studies have used traditional machine learning models to fit the survival of patients with certain diseases. These machine learning models can play a role in predicting patient prognosis, but they are linear models. In real life, the relationship between disease characteristics and patient prognosis may be nonlinear. Deep learning algorithms are constructed using a sequence of layers, each consisting of a nonlinear activation function that depends on an unknown vector of weights that is estimated by minimizing a loss function often subject to some regularization [21]. It can discover the nonlinear relationship between features and disease prognosis. The DeepSurv model can also provide personalized treatment recommendations for patients, and based on the recommendations, it can maximize the patient’s lifespan. For H3K27M-DMG, there is currently no deep learning model to predict its prognosis. In this study, we constructed a model to predict the survival of H3K27M-DMG patients through deep learning and recommend individualized treatment. After comparison in this study, the prediction accuracy of the deep survey model is higher than that of traditional machine learning prediction models.

In this study, there were significant differences in age, radiotherapy and ATRT expression between the training and testing sets, as the patients came from two different hospitals. In the presence of these differences, the DeepSurv model demonstrated strong generalization performance with a prediction accuracy of 81.1% in the test set. In our study, the average survival times of patients in the training and testing sets were 9.41 ± 12.11 years and 10.37 ± 8.78 years, respectively, which is consistent with the previously reported overall survival time of approximately 12 months [22,23]. The disease progression is rapid, so treatment should be carried out as early as possible.

In our study, it was found that the extent of surgical resection was not a significant prognostic factor. Previous studies on glioma survival believed that the more tumors are removed, the longer the overall survival of patients [24,25,26]. The reason why our results differ from theirs may be that they studied low-grade gliomas with low malignancy, whereas H3K27M-DMG is highly aggressive. Taking DIPG as an example, autopsy case statistics show that although the tumor originates in the pons, it can extensively invade areas such as the midbrain and medulla oblongata, forming subclinical infiltrating lesions composed of tumor cells in these areas [27]. However, the extent of surgical resection often does not include this area, and the remaining tumor cells are prone to rapid recurrence.

In our study, KPS played the most important role in the DeepSurv and RSF models. The lower the KPS was, the higher the risk of death for patients, as a lower KPS often indicates that the tumor has progressed to advanced stages. The association of KPS with the risk of death in glioma patients has been demonstrated in numerous studies. Haley et al. [28] plotted a nomogram of low-grade gliomas, which showed that a lower KPS was associated with a higher risk of death in patients. Bai et al. [29] also found that a lower KPS was associated with a higher mortality risk in patients. In this study, tumor size, tumor location, enhancement, radiotherapy, and chemotherapy were also significant prognostic factors. Previous studies have found that tumor location [30,31], radiotherapy [12,31], and chemotherapy [32] are significant prognostic factors for H3K27M-DMG. In the future, larger cohort studies are needed to explore whether tumor size and enhancement are significant prognostic factors for H3K27M-DMG.

Koji et al. [33] compared the survival outcome prediction of the Cox model and deep learning model in clinical cancer. They found that deep learning models outperformed COX models in predicting progression-free survival and overall survival, and as input features increased, the predictive performance of deep learning models further increased. Shreyesh et al. [34] used deep learning to predict the survival of lung cancer patients and found that the deep learning models outperformed traditional machine learning models across both classification and regression approaches. However, there is currently no research using deep learning algorithms to predict the survival of H3K27M-DMG. A previous study used Cox proportional hazard regression for survival analysis of H3K27M-DMG and produced a nomogram [35]. Compared to their study, the advantage of our study is that it adopted deep learning algorithms, included more patients, and additionally collected tumor size information. Therefore, in terms of model performance, our prediction accuracy is higher (c-index, 81.1% vs. 78.5%). In our research, the deep survey model not only outperformed traditional machine learning models in both training and validation sets, but also had the ability to provide personalized recommendations for the postoperative treatment of patients. Not every H3K27M-DMG patient can benefit from adjuvant radiotherapy and chemotherapy after surgery [36]. Therefore, with increasing emphasis on individualized treatment, our model may provide a reference for patients in deciding whether to choose adjuvant radiotherapy and chemotherapy after surgery.

There are some limitations to this study. First, the number of patients in this study was relatively small, and the performance of the model may be improved in future training with a larger number of patients. Second, this study is a retrospective study, and some clinical data of patients are already missing. Third, the failure to incorporate more modal data into the survival model, such as omics, might improve its performance. It is expected that future studies can improve the number of patients and types of input characteristics to further improve the performance of the model.

## 5. Conclusions

We constructed traditional machine learning survival models and the DeepSurv survival prediction model for H3K27M-DMG. The c-indexes of the DeepSurv model in the training and testing sets were 86.2% and 81.1%, respectively. After comparison, the DeepSurv model outperforms traditional machine learning models in terms of prediction accuracy and robustness. The DeepSurv model may provide decision-making assistance for patients in establishing clinic treatment programs in the future. Due to the rarity of H3K27M-DMG, multicentric studies with larger sample size are needed to validate and optimize the model in future work.

## Figures and Tables

**Figure 1 brainsci-13-01483-f001:**
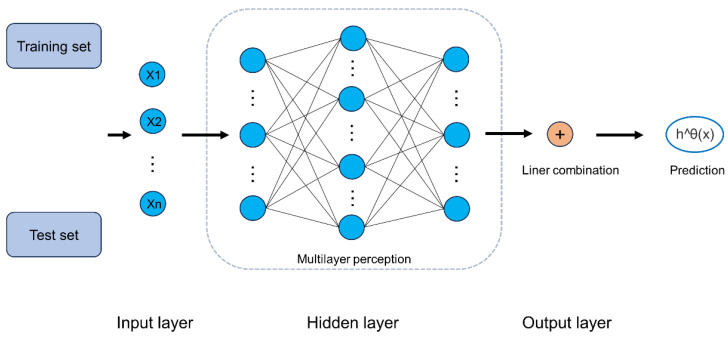
Diagram of DeepSurv. The input to the network is the baseline data x. The network propagates the inputs through a number of hidden layers with weights θ. The hidden layers consist of fully connected nonlinear activation functions followed by dropout. The final layer is a single node which performs a linear combination of the hidden features.

**Figure 2 brainsci-13-01483-f002:**
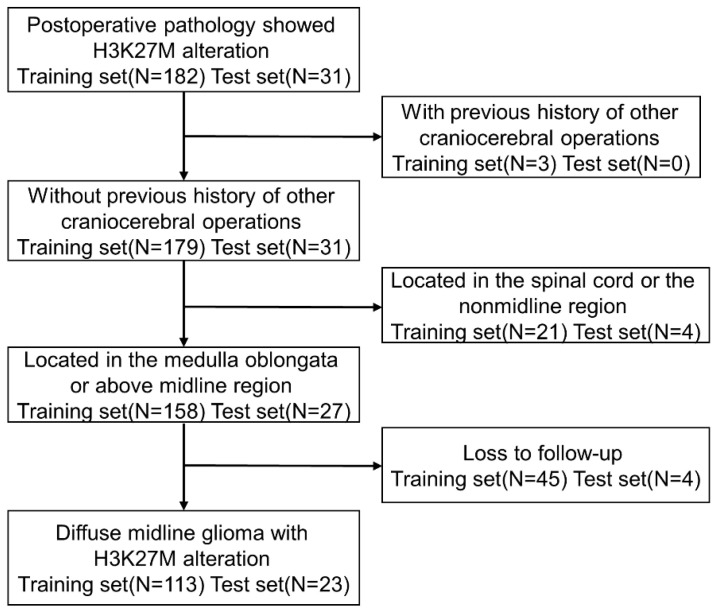
Workflow of the study population.

**Figure 3 brainsci-13-01483-f003:**
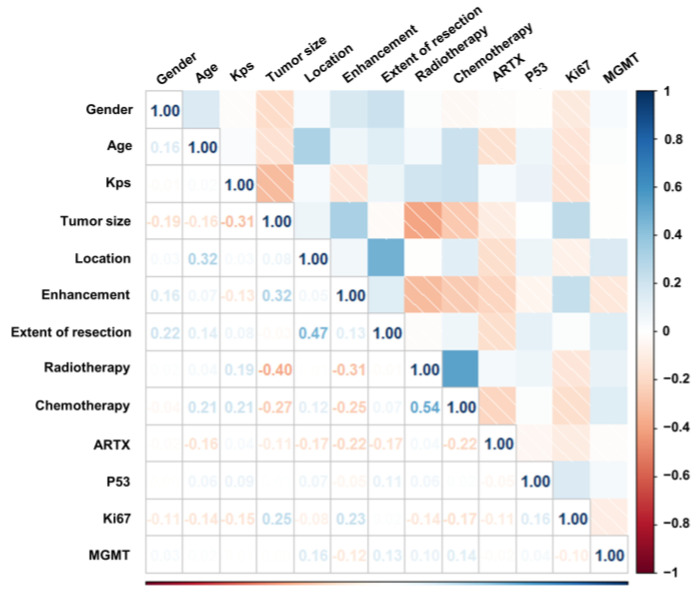
Correlogram illustrating the correlation between all variables. The correlation coefficient is distributed in the range of −1 to +1. They are represented by color depth, and the closer the numbers are to the final value, the stronger their negative or positive correlation.

**Figure 4 brainsci-13-01483-f004:**
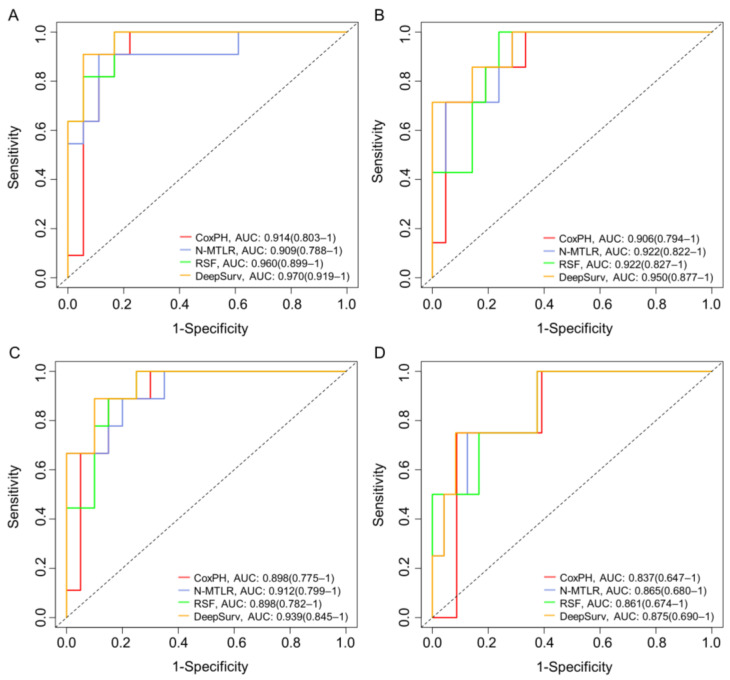
The receiver operating curves (ROC) for 6 (**A**), 12 (**B**), 18 (**C**), and 24 (**D**) months survival predictions.

**Figure 5 brainsci-13-01483-f005:**
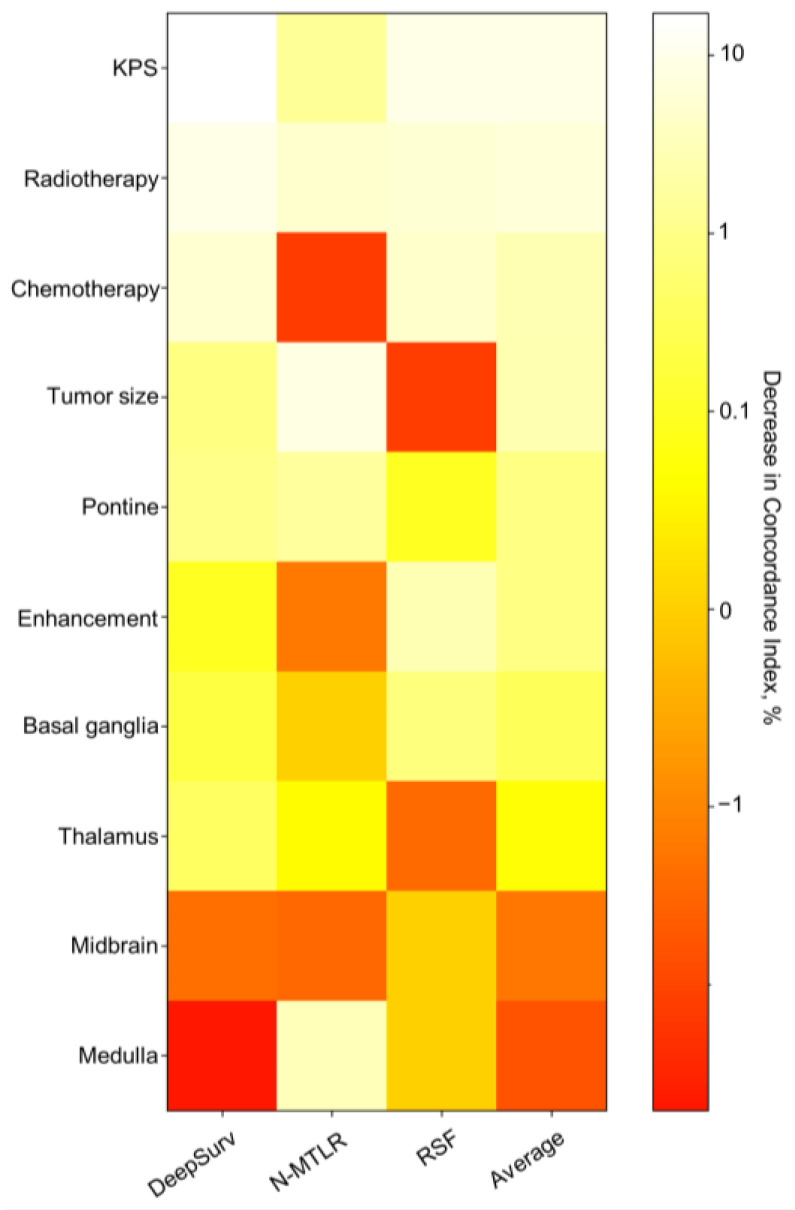
Heatmap of feature importance for DeepSurv, neural network multitask logistic regression (N-MLTR), and random survival forest (RSF) models. Values are given as the percentage decrease in the C index. Higher values indicate greater importance to the predictive accuracy of the respective deep learning model.

**Table 1 brainsci-13-01483-t001:** Univariate and multivariate Cox proportional hazard regression analyses to determine prognostic factors for patients with H3K27M-DMG.

Variable	Univariate Analysis (HR, 95% CI)	*p*	Multivariate Analysis (HR, 95% CI)	*p*
Age	0.988 (0.976–1.000)	0.059	0.994 (0.940–0.970)	0.434
Gender		0.674		0.890
Female	1 [Reference]		1 [Reference]	
Male	0.919 (0.620–1.362)		1.036 (0.627–1.713)	
Tumor size		0.000		0.035
≥1 mm	1 [Reference]		1 [Reference]	
≥2 mm	2.027 (0.728–5.644)		4.069 (1.038–15.958)	
≥3 mm	4.946 (2.069–11.825)		4.848 (1.413–16.637)	
≥4 mm	8.536 (3.504–20.793)		6.771 (1.875–24.457)	
Tumor location		0.037		0.010
Medulla	1 [Reference]		1 [Reference]	
Pontine	0.981 (0.641–5.161)		0.959 (0.145–6.347)	
Midbrain	0.733 (0.103–0.723)		0.050 (0.03–0.718)	
Thalamus	0.801 (0.457–3.715)		0.527 (0.082–3.402)	
Basal ganglia	0.942 (0.445–6.867)		0.801 (0.117–5.487)	
Extent of resection		0.432		0.245
Biopsy	1 [Reference]		1 [Reference]	
PR	1.035 (0.488–2.196)		0.489 (0.200–1.191)	
STR	0.694 (0.308–1.562)		0.404 (0.158–1.038)	
GTR	1.031 (0.455–2.336)		0.598 (0.217–1.651)	
Pre-op KPS	0.964 (0.952–0.975)	0.000	0.955 (0.940–0.970)	0.000
Enhancement		0.000		0.031
No	1 [Reference]		1 [Reference]	
Yes	2.212 (1.462–3.347)		1.733 (1.051–2.859)	
Radiotherapy		0.000		0.000
No	1 [Reference]		1 [Reference]	
Yes	0.203 (0.121–0.342)		0.178 (0.089–0.355)	
Chemotherapy		0.000		0.002
No	1 [Reference]		1 [Reference]	
Yes	0.240 (0.146–0.395)		0.345 (0.175–0.681)	
ATRX expression		0.845		0.112
No	1 [Reference]		1 [Reference]	
Yes	1.044 (0.674–1.617)		1.586 (0.897–2.805)	
P53 positive		0.572		0.066
No	1 [Reference]		1 [Reference]	
Yes	0.858 (0.508–1.448)		0.567 (0.309–1.039)	
Ki67 expression	10.186 (2.735–37.942)	0.001	2.533 (0.511–12.565)	0.255
MGMT promoter methylation		0.193		0.564
Unmethylated	1 [Reference]		1 [Reference]	
Methylated	0.713 (0.421–1.208)		1.200 (0.647–2.225)	

**Table 2 brainsci-13-01483-t002:** The performance of the models in the two datasets.

Models		C-Index	IBS	6 Months AUC	12 Months AUC	18 Months AUC	24 Months AUC
CoxPH	Training set	0.819	0.126	0.914 (0.803–1)	0.906 (0.794–1)	0.898 (0.775–1)	0.837 (0.647–1)
	Test set	0.751	0.162	0.853 (0.781–0.952)	0.836 (0.725–0.947)	0.829 (0.711–0.924)	0.773 (0.607–0.891)
N-MTLR	Training set	0.824	0.104	0.909 (0.788–1)	0.922 (0.822–1)	0.912 (0.799–1)	0.865 (0.680–1)
	Test set	0.763	0.159	0.849 (0.742–0.957)	0.853 (0.765–0.972)	0.849 (0.762–0.974)	0.807 (0.653–1)
RSF	Training set	0.845	0.112	0.960 (0.899–1)	0.922 (0.827–1)	0.898 (0.782–1)	0.861 (0.674–1)
	Test set	0.786	0.150	0.871 (0.805–0.962)	0.853 (0.761–0.985)	0.821 (0.726–0.947)	0.780 (0.637–1)
DeepSurv	Training set	**0.862**	**0.093**	**0.970 (0.919–1)**	**0.950 (0.877–1)**	**0.939 (0.845–1)**	**0.875 (0.690–1)**
	Test set	**0.811**	**0.147**	**0.893 (0.827–0.972)**	**0.869 (0.782–0.961)**	**0.866 (0.776–0.962)**	**0.803 (0.667–1)**

Note: Bolded values indicate that the value is the best of the four groups. Abbreviations: IBS, Integrated Brier Score. CoxPH, Cox proportional hazard model. N-MTLR, NeuralMultiTask Logistic Regression model. RSF, Random Survival Forest model.

## Data Availability

The datasets used and analyzed during the current study are available from the corresponding author on reasonable request.

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
