# Peer review of "Deep Learning for the Prediction of the Survival of Midline Diffuse Glioma with an H3K27M Alteration"

_brainsci, 2023, doi:10.3390/brainsci13101483_

Round 1
Reviewer 1 Report
1. The contributions can be highlighted at the end of the introduction
2. Give a figure label bottom of the represented image. Further, note that figure 1 label is not provided.
3. Provide few more specification details on architecture used
4. Discuss the prognostic factors / their significance in the proposed study for patients with H3K27M-DMG
5. Clearly indicate the permanence measures provided in the table2. The performance parameters such as accuracy/sensitivity...etc are required to be represented in tabel
6. Figure 6, menu screenshot can be removed. Further, explain the significance of the curve given
7. Compare the proposed method's performance with any similar work
8. The conclusion must include some highlights of the proposed study with the outcome/performance. Further, include possible future research.
Reviewer 2 Report
The study is very interesting to read and has been conducted in a scientific manner. However, there are certain concerns as mentioned below. It is highly recommended to incorporate them to improve the paper quality.
1. Title: The title can be changed to "not" include the technical term "H3K27M"
2. The reasons behind the choice of ML/DL algorithms is required. Why have the authors chosen these models for the experiments?
3. Line 67 and 117: The mention of DeepSurv model in the study requires clarity. Is this a standard model which is available for use? Or the authors have developed this model? If it is standard then cite it in reference section.
4. Line 90: Include a sample of the dataset to show the possible values it can take. Show both the training and testing dataset samples.
5. Figure numbers and titles: Usually they are mentioned below the figure. Cross check and confirm with the paper template/format of the journal.
6. Line 132: Explain the reason for omitting the validation set in your experiments.
7. Line 164 and Line 174: The figure are referred in inconsistent manner (Figure 2 and Fig 2). Fix all the occurrences.
8. Line 193: Is it a typo? (Figure S2). There is no such figure in the paper.
9. After line 194: Why a white space? Delete all white spaces by adjusting the text, figures and table. Recommended to use LaTeX, rather than Word.
10. Line 197: Is the Figure 3, described in the main body of the paper? I might have missed it. Kindly highlight or add description.
11. Line 200: Adjust the contents of Table 1, so that it fits in one page.
12. Line 206: Table S2-S3, is it a typo? If not, it should be included in the main paper, rather than a supplementary material.
13. Line 214: Explain the reasoning behind the decrease in the AUC values. Also, tell its impact on the results and related conclusions.
14. Figure 4: Center align it.
15. Figure sizes: They are too large and hence are creating "lots" of white spaces which makes the paper appear less presentable. Implement proper formatting.
16. Figure 6: I would recommend a better resolution which will make the figure more relevant to the readers understanding.
17. New section: Include a new section before "Discussion" as "Related Work" which describes the relevant work in this domain. Summarize this new section with a table having these columns (Year, Approach, Dataset, Results, Pros and Cons)
18. Another table in Discussions section: Include a new table which provides the comparative study with your results. It should include these columns (Year, Approach, Results)
19. References: Shorten references 21 and 22 by either including max 5 authors or using et. al.
20. Run a spell check on the document before re-submitting.
The English is very much readable and understandable. However, it is recommended to run a spell check and proof reading by a native English speaker.
Reviewer 3 Report
The manuscript by Bowen Huang and colleagues presents a comprehensive exploration into the prognosis of diffuse midline glioma (DMG) patients with H3K27M alterations. The primary objective is to construct a survival model based on DeepSurv to predict patient prognosis. The study is methodologically robust, utilizing distinct patient cohorts for model training and external validation. The DeepSurv model, among the four machine learning models constructed, demonstrated superior accuracy and robustness. Additionally, the introduction of an interactive interface to display survival probability prediction results is a commendable initiative.
Minor comments:
1. Figure 4: The absence of statistical data in Figure 4 is a notable omission. Including this would enhance the comprehensiveness of the results.
2. The introductory section could be enriched by incorporating recent research pertinent to the study, providing readers with a more comprehensive background.
3. Some sections of the manuscript, particularly certain phrases, are somewhat challenging to interpret. A thorough review by a native English speaker is recommended to ensure clarity and coherence.
Some sections of the manuscript, particularly certain phrases, are somewhat challenging to interpret. A thorough review by a native English speaker is recommended to ensure clarity and coherence.
Round 2
Reviewer 2 Report
The authors have improved the paper quality, but have to take care of minor issues as below:
1. This reference HAS to be included in the reference section, not just in the reply to reviewer comments:
Reference:1、Kiessling J, Brunnberg A, Holte G, Eldrup N, Sörelius K. Artificial Intelligence Outperforms Kaplan-Meier Analyses Estimating Survival after Elective Treatment of Abdominal Aortic Aneurysms. European journal of vascular and endovascular surgery : the official journal of the European Society for Vascular Surgery. 2023;65(4):600-607.
2. The following text does not refer to Figure 3. It SHOULD be included as part of the updated manuscript at this location.
(We use the cor function in R software to calculate the interrelationships between these features and test whether there is collinearity between them. When Pearson's correlation value is ≥ 0.7, it means that these factors have a high degree of collinearity.)
